# Ultra-processed food intake in association with BMI change and risk of overweight and obesity: A prospective analysis of the French NutriNet-Santé cohort

Marie Beslay[1‡], Bernard Srour[1‡*], Caroline Méjean[2], Benjamin Allès[1], Thibault Fiolet[1], Charlotte Debras[1], Eloi Chazelas[1], Mélanie Deschasaux[1], Méyomo Gaelle Wendeu-Foyet[1], Serge Hercberg[1,3], Pilar Galan[1], Carlos A. Monteiro[4], Valérie Deschamps[5], Giovanna Calixto Andrade[1,6], Emmanuelle Kesse-Guyot[1], Chantal Julia[1,3], Mathilde Touvier[1]

**1** Sorbonne Paris Nord University, Inserm, INRAE, Cnam, Nutritional Epidemiology Research Team (EREN), Epidemiology and Statistics Research Center–University of Paris (CRESS), Bobigny, France, **2** MOISA, Univ Montpellier, CIRAD, CIHEAM-IAMM, INRAE, Montpellier SupAgro, Montpellier, France, **3** Public Health Department, Avicenne Hospital, AP-HP, Bobigny, France, **4** Department of Nutrition, School of Public Health, University of São Paulo, São Paulo, Brazil, **5** Santé Publique France (The French Public Health Agency), Nutritional Epidemiology Surveillance Team (ESEN), **6** Department of Preventive Medicine, Medical School, University of São Paulo, São Paulo, Brazil

‡These authors are co-first authors and contributed equally to this work. CJ and MT are co-last authors and contributed equally to this work.

\* b.srour@eren.smbh.univ-paris13.fr

**Data Availability Statement:** Data of the study are protected under the protection of health data regulation set by the French National Commission

## Abstract

### Background

Ultra-processed food (UPF) consumption has increased drastically worldwide and already represents 50%–60% of total daily energy intake in several high-income countries. In the meantime, the prevalence of overweight and obesity has risen continuously during the last century. The objective of this study was to investigate the associations between UPF consumption and the risk of overweight and obesity, as well as change in body mass index (BMI), in a large French cohort.

### Methods and findings

A total of 110,260 adult participants (≥18 years old, mean baseline age = 43.1 [SD 14.6] years; 78.2% women) from the French prospective population-based NutriNet-Santé cohort (2009–2019) were included. Dietary intakes were collected at baseline using repeated and validated 24-hour dietary records linked to a food composition database that included >3,500 different food items, each categorized according to their degree of processing by the NOVA classification. Associations between the proportion of UPF in the diet and BMI change during follow-up were assessed using linear mixed models. Associations with risk of overweight and obesity were assessed using Cox proportional hazard models. After adjusting for age, sex, educational level, marital status, physical activity, smoking status, alcohol intake, number of 24-hour dietary records, and energy intake, we observed a positive

on Informatics and Liberty (Commission Nationale de l'Informatique et des Libertés, CNIL). The data can be available upon request to Nathalie Pecollo (n.pecollo@eren.smbh.univ-paris13.fr), after a consultation with the steering committee of the NutriNet-Santé study. The French law forbids us to provide free access to NutriNet-Santé data; access could be exceptionally granted by the steering committee after legal verification of the use of the data.

**Funding:** NutriNet-Santé was supported by the following public institutions: Ministère de la Santé (solidarites-sante.gouv.fr), Santé Publique France (santepubliquefrance.fr), Institut National de la Santé et de la Recherche Médicale (INSERM) (inserm.fr), Institut National de la Recherche Agronomique (INRAE) (inrae.fr), Conservatoire National des Arts et Métiers (CNAM) (cnam.fr), and Université Sorbonne Paris Nord (univ-paris13.fr). EC was supported by a Doctoral Funding from Université Sorbonne Paris Nord - Galilée Doctoral School (univ-paris13.fr). MD was supported by a grant from the Fondation pour la Recherche Médicale (frm.org). CD was supported by a grant from the French National Cancer Institute (INCa) (e-cancer.fr). Researchers were independent from funders. Funders had no role in the study design, collection, analysis, and interpretation of data, the writing of the report, and the decision to submit the article for publication.

**Competing interests:** The authors have declared that no competing interests exist.

**Abbreviations:** BMI, body mass index; CNIL, French National Commission on Informatics and Liberty; HR, hazard ratio; ICD-10, Tenth Revision of the International Statistical Classification of Diseases and Related Health Problems; MICE, Multiple Imputation by Chained Equations; PNNS, French National Programme for Nutrition and Health; SFA, saturated fatty acid; STROBE, Strengthening the Reporting of Observational Studies in Epidemiology; UPF, ultra-processed food; WCRF/AICR, World Cancer Research Fund/ American Institute for Cancer Research.

association between UPF intake and gain in BMI ($\beta$ Time × UPF = 0.02 for an absolute increment of 10 in the percentage of UPF in the diet, $P < 0.001$). UPF intake was associated with a higher risk of overweight ($n = 7,063$ overweight participants; hazard ratio (HR) for an absolute increase of 10% of UPFs in the diet = 1.11, 95% CI: 1.08–1.14; $P < 0.001$) and obesity ($n = 3,066$ incident obese participants; $HR_{10\%} = 1.09$ (1.05–1.13); $P < 0.001$). These results remained statistically significant after adjustment for the nutritional quality of the diet and energy intake. Study limitations include possible selection bias, potential residual confounding due to the observational design, and a possible item misclassification according to the level of processing. Nonetheless, robustness was tested and verified using a large panel of sensitivity analyses.

## Conclusions

In this large observational prospective study, higher consumption of UPF was associated with gain in BMI and higher risks of overweight and obesity. Public health authorities in several countries recently started to recommend privileging unprocessed/minimally processed foods and limiting UPF consumption.

## Trial registration

ClinicalTrials.gov NCT03335644 (https://clinicaltrials.gov/ct2/show/NCT03335644)

## Author summary

### Why was this study done?

- Ultra-processed food (UPF) consumption has increased drastically worldwide and already represents 50%–60% of total daily energy intake in several high-income countries.

- These changes in dietary behaviours are concomitant with a continuous rise in the prevalence of overweight and obesity during the last century.

- Only 2 prospective studies investigated the associations between UPF consumption and overweight/obesity risks.

- We studied the associations between the contribution of UPF to the diets of more than 100,000 participants and longitudinal changes in body mass index (BMI), as well as risks of overweight and obesity.

### What did the researchers do and find?

- We used appropriate statistical models to study the associations between UPF consumption and BMI change, as well as risks of overweight and obesity, in the NutriNet-Santé cohort.

- 110,260 participants were followed between 2009 and 2019.

- Having a higher consumption of UPF was associated with an increased weight gain, as well as increased risks of becoming overweight or obese.

- These associations were not fully explained by the overall poorer nutritional quality of UPF.

### What do these findings mean?

- These results suggest that consumption of UPF is associated with increased weight gain.

- This study contributes to the mounting evidence on the link between food processing and health.

- Further studies (epidemiological and experimental) are needed to investigate the relative contribution of nutritional composition, food additives, process- or packaging-related contaminants, and modification of the food matrix.

- UPF consumption should be limited, and the consumption of unprocessed or minimally processed foods should be promoted instead, as several national nutritional policies recommend.

## Introduction

Obesity and overweight nowadays affect a large share of the world's population: in 2016, 13% of adults aged over 18 were estimated to be obese and 39% to be overweight [1]. The prevalence of overweight and obesity has risen continuously during the last century, in particular in low-income countries as well as in low-income groups in high-income countries, in both adults and children [2]. In France, almost half of adults were overweight or obese and 17% were obese in 2015 [3]. Obesity is not only a major risk factor of metabolic diseases (such as coronary heart disease, ischemic stroke, type 2 diabetes [1]) and cancer [4,5], but it is also a metabolic disease itself (Tenth Revision of the International Statistical Classification of Diseases and Related Health Problems [ICD-10] code E66). The physiological and psychological consequences of obesity also significantly impair the quality of life and constitute a further burden for health systems [6].

Prevention of chronic diseases has therefore been considered a public health challenge in the past decades [2]. Besides physical activity, the nutritional quality of the diet is a major modifiable risk factor for weight management, with strong levels of evidence for protective factors (i.e., dietary fibre, Mediterranean diets) and risk factors (i.e., high energy density, free sugars, sugar sweetened drinks, and Western-type diets) [5,7]. In addition, drivers of the global obesity epidemic might reside in the change in social behaviours and environmental factors (such as built environment) [8–10]. Changes in the food system are likely to play a key role in the obesity pandemic [11]: they are notably characterized by increased supply of affordable, hyperpalatable energy-dense food products, along with sophisticated distribution systems to improve accessibility and convenience and intensive food marketing campaigns. These trends in the food systems were accompanied by major dietary changes in the last decades. In particular, industrially processed products and especially ultra-processed food (UPF) consumption drastically rose [12] representing already 50%–60% of total daily energy in some high-income

countries [13–16]. UPFs have a poorer nutritional quality (often high in energy, salt, free sugars, and saturated fats and low in fibre and vitamins [13–15,17–24]) compared to unprocessed food. Another characteristic of UPFs is that they are obtained after sequences of several processes, such as high-temperature extrusion, moulding, and pre-frying, and include several food additives and industrial ingredients used to imitate or enhance sensory qualities of foods or to disguise unpalatable aspects of the final product; they are also often in contact with synthetic packaging materials for long periods [25].

Recent evidence suggests adverse associations between UPF consumption and several chronic diseases [26], including studies conducted in the framework of the NutriNet-Santé cohort that have shown positive associations with risks of cancer [27], cardiovascular diseases [24], depressive symptoms [28], type 2 diabetes [29], and all-cause mortality [30]. Regarding weight change and obesity (a risk factor for the latter chronic diseases [5,31–33]), a 2-week randomized cross-over trial [34] showed that an ultra-processed diet versus an unprocessed one led to an increased daily energy intake of around 500 kcal which was highly correlated with weight gain. Consistently, several cross-sectional and ecological studies have substantiated a positive association between UPF consumption and obesity [23,35–39], but prospective studies are lacking, as only 2 of them—one in Spain [40] and one in Brazil [41]—were conducted; both relied on dietary data from food frequency questionnaires (FFQs) and studied the contribution of UPF to energy intake and therefore did not account for low-calorie and artificially sweetened products.

This large-scale prospective study aimed to investigate the associations between UPF consumption and body mass index (BMI) change, as well as the risk of overweight and obesity, among adults from the large-scale NutriNet-Santé cohort.

## Methods

### Population

The NutriNet-Santé study is a French web-based ongoing cohort study, launched in 2009 to investigate the associations between diet and health. The NutriNet-Santé cohort has been previously described in detail [42]. Briefly, participants from all regions of France with access to the internet have been continuously recruited, on a voluntary basis, from the general population since May 2009. Vast multimedia campaigns (television, radio, national and regional newspapers, posters, internet) called for volunteers by providing details on the study's specific website where volunteers can subscribe. A relay of information was also maintained on a large number of websites (national institutions, city councils, private firms, web organisations). A billboard advertising campaign was also available through professional channels (doctors, pharmacists, dentists, business partners, municipalities, etc.) [43]. The online NutriNet-Santé platform is designed to send an average of 1 questionnaire per month, allowing us to collect additional information on various research topics beyond diet (e.g., sleep duration, environmental exposures, mental health, cooking practices). The NutriNet-Santé study was conducted in accordance with the Declaration of Helsinki and was approved by the ethics committee of the French Institute for Health and Medical Research (IRB Inserm No. 0000388FWA00005831) and by the National Commission on Informatics and Liberty (CNIL No. 908450 and No. 909216). The study is registered at clinicaltrials.gov as NCT03335644. Some other information can be accessed on the website https://info.etude-nutrinet-sante.fr/en. Electronic informed consent was obtained from each participant. All methods have been described in line with the Strengthening the Reporting of Observational Studies in Epidemiology (STROBE) Statement (see S1 STROBE Checklist).

## Data collection

**Dietary data.** Dietary data were collected at baseline using a kit of 3 non-consecutive web-based 24-hour records, randomly assigned over a 2-week period (2 during weekdays and 1 during the weekend). Participants were invited to declare every beverage and food consumed that day, during the 3 main meals and any additional eating occasions. Portion sizes were assessed using validated photographs or usual containers. In this prospective analysis, we averaged the mean dietary intakes from the baseline 24-hour dietary records and considered these as baseline usual dietary intakes. Nutrient intakes were calculated using a food composition table, listing more than 3,500 food items [44]. The contribution of macronutrients to total energy intake was calculated. Dietary underreporting was identified with the method proposed by Black, using the basal metabolic rate and Goldberg cut-off, in order to screen participants with abnormally low energy intakes, and energy underreporters (20.0% of the cohort) were excluded [45]. Detailed methodology for underreporting is presented in Method A in S1 Appendix. Validation studies comparing these web-based dietary questionnaires to interviews by dieticians [46] or urinary and plasma biomarkers [47,48] of nutritional status demonstrated a good validity of the collected data.

**Food processing classification.** Foods and beverages of the NutriNet-Santé composition table were categorized according to the extent of processing, into one of the 4 NOVA categories (unprocessed/minimally processed foods, processed culinary ingredients, processed foods, UPFs) [25,49]. This categorization was performed by a team of 3 dietitians and 5 researchers [39]. Home-made and artisanal foods were identified and decomposed using standardized recipes, and the classification was applied to their ingredients. In case of uncertainty, classification was based on the consensus reached in the team. Details and examples are provided in Method B in S1 Appendix.

**Anthropometric data.** Self-reported weight and height were collected using a web-based questionnaire at baseline, and every 6 months thereafter between May 2009 and June 2019, and were used to compute repeated data of BMI (BMI = (weight [kilograms] ÷ height$^2$ [meters]). Obesity was identified using international standards as a BMI $\geq$ 30 kg/m$^2$ and overweight including obesity was identified as a BMI $\geq$ 25 kg/m$^2$ [2]. This web-based questionnaire was validated by comparison with standardized clinical measurements [50].

**Covariates.** Sociodemographic data were collected at baseline using a self-administered questionnaire [51]. Sex, age, educational level (no higher education, <2 years after high school, $\geq$2 years after high school), marital status (living alone or not), and smoking status (current, former, or never smoker) were collected for each participant. Physical activity was computed using the validated International Physical Activity Questionnaire, completed at baseline (low, moderate, and high physical activity levels) [52] (details in Method C in S1 Appendix). Data about time spent for screen watching and sedentary behaviours were also available. Several indicators of the nutritional quality of the diet were also computed based on average dietary intakes from baseline 24-hour dietary records and were used as covariates: daily nutrient intake (sugar, fibre, sodium, and saturated fatty acid [SFA]) calculated using the food composition database; consumptions of several food groups (fruit, vegetables, and sugary drinks); and healthy and Western dietary patterns, derived from Principal Component Analysis (see Method D in S1 Appendix).

## Statistical analysis

Analyses for this specific article were hypothesis oriented in order to investigate the relationship between UPF consumption and weight gain or prospective occurrence of overweight or obesity. No specific analysis plan has been pre-published for the present article, but all analyses

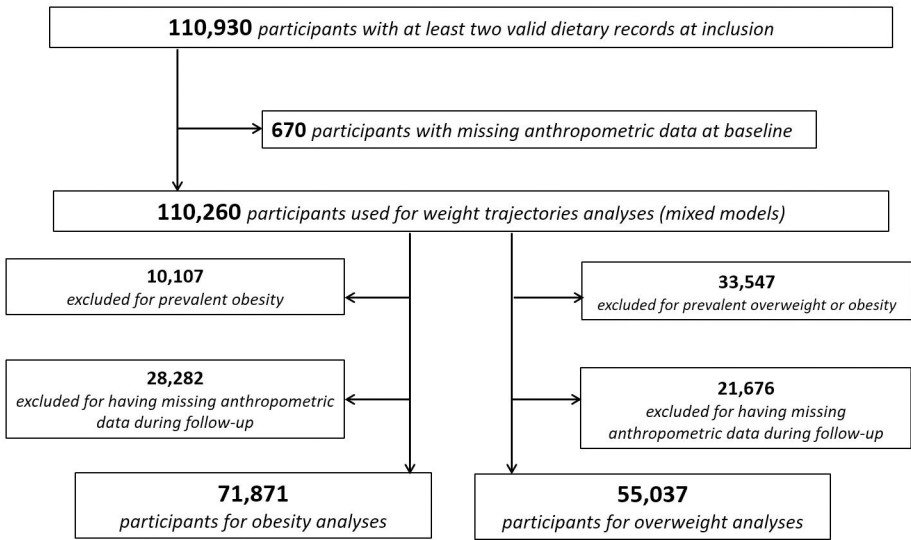

**Fig 1. Flowchart for study population, NutriNet-Santé cohort, 2009–2019.**

were pre-planned by the authors at the time of conception and design of the present study, except one non-prespecified analysis that was performed to comply with peer-review requirements (mixed models with continuous exposure).

Adults (aged between 18.0 and 73.3 years old) who completed at least two 24-hour records and who were followed up for at least 6 months, with no missing anthropometric data at baseline and with at least 2 available anthropometric questionnaires, were included. Details are shown in the flowchart (Fig 1).

For each participant, the proportion (in weight, percentage of grams per day) of UPF in the total diet was calculated. The proportion of UPF in the diet was determined with a weight ratio rather than an energy ratio in order to better take into account food items that do not provide any energy (e.g., artificially sweetened beverages) and non-nutritional issues related to food processing (e.g., food additives, neo-formed contaminants and alterations to the structure of raw foods).

The population's characteristics were described according to sex-specific quartiles of the proportion of UPF in the diet (quartiles were built separately in men and women according to the specific distribution and cut-offs in each group; matching quartiles were then combined). For all covariates except physical activity, ≤5% of values were missing and imputed to the modal (categorical variables) or median (continuous variables) values. A missing class was created for physical activity (14% missing). Multiple Imputation by Chained Equations (MICE) method was also tested [53].

We measured the associations between the proportion of UPF in the diet (for an absolute increment of 10 in the percentage of UPF in the diet coded as a continuous variable, and as sex-specific quartiles) and BMI change over time using mixed models for repeated measures (PROC MIXED in SAS), with UPF as fixed effect and intercept and time as random effects, with unstructured covariance structure. Time was defined as the chronological number of the anthropometric questionnaire (approximately 1 year of follow-up = 2 time units), from which the corresponding data were collected. The outcome modelled was the absolute change in BMI (Δ BMI). Models were adjusted for age, sex, educational level, smoking status, marital status, physical activity level, energy intake, alcohol intake, and number of dietary records. Additional adjustments were tested in different models: for sugar (grams per day), fibre (grams per day),

sodium (grams per day), and SFA intake (grams per day) (to adjust for the nutritional quality of UPF); for dietary patterns (to capture the overall quality of the diet); and for consumptions of fruit, vegetables (grams per day), and sugary drinks (millilitres per day) (convincingly linked to the risk of weight gain according to the World Cancer Research Fund/American Institute for Cancer Research [WCRF/AICR]) [5].

Associations between quartiles of UPF consumption and overweight and obesity risk were assessed using Cox proportional hazard models with age as timescale. Schoenfeld residuals were generated to confirm the risk proportionality assumptions. Martingale residuals were generated to confirm the assumption of linearity for the percentage UPF in the diet when used as continuous. Participants contributed person-time to the Cox model until the date of onset of overweight or obesity for cases (defined as the mid-point date between the anthropometrics questionnaire in which the participant's self-reported weight corresponding to overweight or obesity, and the previous one [54]) and the date of last completed anthropometrics question-naire for non-cases. Similar adjustments as those used in BMI change analyses were used. Analyses were tested with and without adjustment for baseline BMI.

**Sensitivity and secondary analyses.** Associations were also tested in stratified analyses for sex, age, intake of sugar and SFA, and smoking status, as well as in sensitivity analyses exclud-ing cases occurring within the first 2 years.

In secondary analyses, we explored the associations between the proportion of unpro-cessed/minimally processed foods in the diet (first category of NOVA) in association with the risks of overweight and obesity. We also investigated the associations between the consump-tion amount (in grams per day) of UPF and risks of overweight and obesity, as well as the amounts of the different UPF groups (beverages, dairy products, fats and sauces, fruits and vegetables, meat, fish and egg, starchy foods and breakfast cereals, sugary products, and salty snacks).

Statistical analyses were performed using SAS 9.4 and R Studio. All tests were two-sided, and $P < 0.05$ was considered significant.

## Results

### Descriptive results

Baseline characteristics of the study population ($n = 110,260$, among which women made up 78.2% women; mean age [SD] = 43.1 [14.6]) are described in Table 1. The density plot of the proportion of UPF in the diet in the study sample is presented in Fig A in S1 Appendix. Com-pared to participants with a lower proportion of UPF in their diet (first quartile), participants in the fourth quartile tended to be younger, were more likely to be smokers, were less likely to be single, were higher educated ($\geq$ 2years after high school), and had a lower physical activity level. They had also higher sodium, sugar, SFA, and energy intake and lower intakes of dietary fibre and alcohol.

### UPF and BMI change

BMI change over time by sex-specific quartiles of UPF proportion in the diet is shown in Fig 2, and results of mixed models are presented in Table 2. Participants in the fourth quartile of UPF consumption had higher BMI at baseline (β coefficients for Q4 > 0) compared to those in the first quartile (reference in the model). While an increase of BMI was observed in all UPF quartiles (β coefficients for time significantly > 0), the BMI gain appeared to be higher for par-ticipants in quartiles 2, 3, and 4 compared to individuals from quartile 1 (β coefficients for interactions terms between time and quartile > 0); the magnitude of BMI increase was the highest for Q4 (βQ4 × time = 0.04 [0.04–0.05], $P < 0.001$, model 1). In continuous models, we

**Table 1. Baseline characteristics of the study population according to sex-specific quartiles of UPF consumption (N = 110,260), NutriNet-Santé cohort, France, 2009–2019[a].**

| | All participants | Quartile[b] 1 | Quartile[b] 2 | Quartile[b] 3 | Quartile[b] 4 |
|---|---|---|---|---|---|
| | | (n = 27,609) | (n = 27,576) | (n = 27,556) | (n = 27,519) |
| **UPF (%)** | 17.1 (10.3) | 7.5 (2.1) | 13.2 (1.8) | 18.7 (2.2) | 32.4 (9.6) |
| **Age, years** | 43.1 (14.6) | 47.7 (13.7) | 44.9 (14.3) | 42.4 (14.6) | 37.5 (14.1) |
| **Sex, women, n (%)** | 86,253 (78.2) | 21,601 (78.2) | 21,574 (78.2) | 21,553 (78.2) | 21,525 (78.2) |
| **Educational level, n (%)** | | | | | |
| <High school degree | 20,013 (18.1) | 5,212 (18.9) | 4,941 (17.9) | 4,873 (17.7) | 4,987 (18.1) |
| <2 years after high school | 19,061 (17.3) | 4,145 (15.0) | 4,300 (15.6) | 4,764 (17.3) | 5,852 (21.3) |
| ≥2 years after high school | 71,186 (64.6) | 18,252 (66.1) | 18,335 (66.5) | 17,919 (65.0) | 16,680 (60.6) |
| **Marital status, n (%)** | | | | | |
| Single (living alone) | 32,532 (29.5) | 7,380 (26.7) | 7,383 (26.8) | 7,990 (29.0) | 9,779 (35.5) |
| In couple | 77,728 (70.5) | 20,229 (73.3) | 20,193 (73.2) | 19,566 (71.0) | 17,740 (64.5) |
| **BMI, kg/m²** | 23.8 (4.6) | 23.8 ± 4.4 | 23.8 ± 4.3 | 23.8 ± 4.5 | 23.9 ± 5.0 |
| **Smoking status, n (%)** | | | | | |
| Current | 18,731 (17.0) | 4,181 (15.1) | 4,319 (15.7) | 4,526 (16.4) | 5,705 (20.7) |
| Former | 36,243 (32.9) | 10,564 (38.3) | 9,373 (34.0) | 8,852 (32.1) | 7,454 (27.1) |
| Never | 55,286 (50.1) | 12,864 (46.6) | 13,884 (50.3) | 14,178 (51.4) | 14,360 (52.2) |
| **IPAQ physical activity level, n (%)[c]** | | | | | |
| High | 31,638 (33.2) | 9,152 (38.0) | 8,222 (34.0) | 7,671 (32.2) | 6,593 (28.3) |
| Moderate | 40,825 (42.8) | 10,171 (42.2) | 10,516 (43.6) | 10,251 (43.0) | 9,887 (42.4) |
| Low | 22,881 (24.0) | 4,748 (19.7) | 5,391 (22.3) | 5,918 (24.8) | 6,824 (29.3) |
| **Energy intake, kcal/d** | 1,893.2 (503.9) | 1,816.2 (473.8) | 1,896.5 (489.0) | 1,925.5 (507.8) | 1,934.70 (534.5) |
| **Alcohol intake, g/d** | 7.9 (12.7) | 8.8 (13.5) | 8.7 (13.0) | 7.8 (12.4) | 6.2 (11.7) |
| **Sodium intake, mg/d** | 2,683.9 (954.6) | 2,536.2 (919.5) | 2,696.7 (931.5) | 2,762.0 (962.1) | 2,741.0 (987.4) |
| **SFA, g/d** | 32.8 (13.2) | 29.7 (12.2) | 32.7 (12.7) | 33.9 (13.2) | 34.74 (13.9) |
| **Dietary fibre, g/d** | 19.6 (7.6) | 21.0 (8.1) | 20.0 (7.4) | 19.7 (7.3) | 17.7 (7.4) |
| **Sugar, g/d** | 92.2 (35.4) | 85.3 (33.9) | 90.9 (32.7) | 93.7 (34.0) | 99.1 (39.1) |

[a]Values are means (SDs) or n (%).

[b]Sex-specific quartiles of the proportion of UPF intake in the total quantity of food consumed. Cut-offs for quartiles were 10.2, 15.5, and 22.5 for men and 9.9, 15.2, and 22.1 for women, respectively.

[c]Available for 95,344 participants. Participants were categorized into the "high," "moderate," and "low" categories according to IPAQ guidelines [52] (Method C in S1 Appendix).

**Abbreviations:** BMI, body mass index; IPAQ, International Physical Activity Questionnaire; SFA, saturated fatty acid; UPF, ultra-processed food

observed a positive association between an absolute increment of 10 in the percentage of UPF in the diet and gain in BMI (β time × UPF$_{continuous}$ = 0.02 (0.01–0.02), $P < 0.001$, model 1). The findings remained similar after further adjustments for intake of sugar, sodium, SFA, and dietary fibre (model 2), for healthy and Western dietary patterns (model 3), and for intake of fruit and vegetables and sugary drinks (model 4).

## UPF and risk of overweight

Analyses related to overweight (including obesity) risk were performed on a sample of 55,037 non-overweight participants at baseline (Table 3). During follow-up (260,304 person-years, median follow-up time = 4.1 years), 7,063 participants became overweight. The proportional hazard assumptions of the Cox models were met, as well as the linearity assumption for the continuous model (Fig B in S1 Appendix). Participants with a higher proportion of UPF in

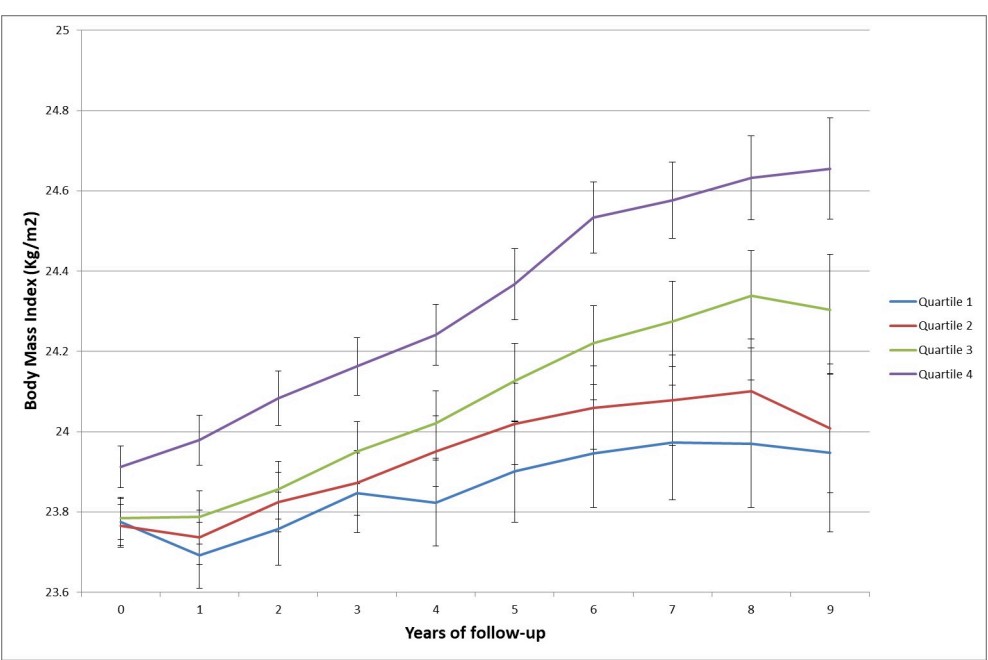

**Fig 2. BMI change over time in the four quartiles of the proportion of UPF in the diet, NutriNet-Santé, 2009–2019 (*n* = 110,260).** The average BMI for each year and each quartile of UPF intake is presented along with the 95% CI of the mean. BMI, body mass index; UPF, ultra-processed food

their diet had a higher risk of becoming overweight (hazard ratio [HR] for an absolute increment of 10 in the percentage of UPF in the diet = 1.11 [1.08–1.14], $P < 0.001$). These trends were significant from the second quartile of UPF intake and were the strongest in the fourth quartile: $HR_{Q4vs.Q1} = 1.26$ (1.18 to 1.35), $P_{trend} < 0.001$. These associations remained significant after adjustment for several indicators of the nutritional quality of the diet.

## UPF and risk of obesity

Analyses related to obesity risk were performed on a sample of 71,871 participants non-obese at baseline (Table 3). During follow-up (365,344 person-years, median follow-up time = 5.0 years), 3,066 participants became obese. The proportional hazard assumptions of the Cox models were met, as well as the linearity assumption for the continuous model (Fig B in S1 Appendix). Participants with a higher proportion of UPF in their diet had a higher risk of obesity (HR for an absolute increment of 10 in the percentage of UPF in the diet = 1.09 [1.05–1.13], $P < 0.001$). These trends were statistically significant starting the third quartile, were the strongest in the fourth quartile ($HR_{Q4vs.Q1} = 1.15$ [1.04–1.28], $P_{trend} = 0.005$), and remained stable across all models with further adjustments.

## Sensitivity and secondary analyses

Using the proportion of UPF weighted by energy (rather than quantity) or the absolute consumption amount (in grams) of UPF did not change the findings (Table A in S1 Appendix). Ultra-processed beverages, dairy products, fats and sauces, and meat, fish, and egg were each associated with increased overweight and obesity risks, while ultra-processed starchy foods and breakfast cereals were associated with an increased risk of overweight but not obesity (Table B in S1 Appendix). In contrast, there was no evidence for a positive association between

**Table 2. Associations between sex-specific quartiles of UPF consumption and BMI change, NutriNet-Santé cohort, France, 2009–2019 (N = 110,260).**

| Proportion of UPF in the diet | Model 1 | | Model 2 | | Model 3 | | Model 4 | |
|---|---|---|---|---|---|---|---|---|
| | β[a] (95% CI) | P | β (95% CI) | P | β (95% CI) | P | β (95% CI) | P |
| **Continuous** | | | | | | | | |
| UPF$_{continuous}$[b] | 0.25 (0.22 to 0.27) | <0.001 | 0.20 (0.17 to 0.21) | <0.001 | 0.14 (0.12 to 0.17) | <0.001 | 0.21 (0.19 to 0.24) | <0.001 |
| Time (average BMI gain/time unit)[c] | 0.02 (0.01 to 0.02) | <0.001 | 0.02 (0.01 to 0.02) | <0.001 | 0.02 (0.01 to 0.02) | <0.001 | 0.02 (0.01 to 0.02) | <0.001 |
| Time × UPF$_{continuous}$[d] | 0.02 (0.01 to 0.02) | <0.001 | 0.02 (0.01 to 0.02) | <0.001 | 0.02 (0.01 to 0.02) | <0.001 | 0.02 (0.01 to 0.02) | <0.001 |
| **Sex-specific quartiles** | | | | | | | | |
| Quartile 2 (BMI difference at baseline with the reference–Q1) | 0.11 (0.04 to 0.19) | 0.003 | 0.02 (-0.05 to 0.09) | 0.6 | −0.00 (−0.07 to 0.07) | 0.9 | 0.04 (−0.03 to 0.12) | 0.2 |
| Quartile 3 (BMI difference at baseline with the reference–Q1) | 0.24 (0.16 to 0.31) | <0.001 | 0.11 (0.03 to 0.18) | 0.004 | 0.05 (−0.02 to 0.13) | 0.1 | 0.13 (0.06 to 0.21) | <0.001 |
| Quartile 4 (BMI difference at baseline with the reference–Q1) | 0.60 (0.52 to 0.67) | <0.001 | 0.42 (0.34 to 0.51) | <0.001 | 0.30 (0.23 to 0.38) | <0.001 | 0.43 (0.35 to 0.52) | <0.001 |
| Time (BMI gain/time unit in the reference–Q1) | 0.03 (0.03 to 0.04) | <0.001 | 0.03 (0.03 to 0.04) | <0.001 | 0.03 (0.03 to 0.04) | <0.001 | 0.03 (0.03 to 0.04) | <0.001 |
| Time × quartile 2 (additional BMI gain/time unit[e] compared to Q1) | 0.01 (0.003 to 0.01) | 0.001 | 0.01 (0.004 to 0.02) | 0.001 | 0.01 (0.003 to 0.02) | 0.001 | 0.01 (0.003 to 0.02) | 0.002 |
| Time × quartile 3 (additional BMI gain/time unit[e] compared to Q1) | 0.02 (0.01 to 0.02) | <0.001 | 0.01 (0.01 to 0.02) | <0.001 | 0.02 (0.01 to 0.02) | <0.001 | 0.02 (0.01 to 0.02) | <0.001 |
| Time × quartile 4 (additional BMI gain/time unit[e] compared to Q1) | 0.04 (0.04 to 0.05) | <0.001 | 0.04 (0.04 to 0.05) | <0.001 | 0.04 (0.04 to 0.05) | <0.001 | 0.04 (0.04 to 0.05) | <0.001 |

Model 1 = mixed model for repeated measure, with intercept and time as random, adjusted for age, sex, marital status (living alone or not), educational level (<high school, <2 years after school, ≥2 years after high school), physical activity level (high, moderate, low), smoking status (never, former, current), alcohol consumption (continuous), energy intake (continuous), and number of dietary records (continuous); model 2 = model 1 + intakes of sugar, sodium, SFAs, and dietary fibre (continuous); model 3 = model 1 + healthy and Western dietary patterns (continuous); model 4 = model 1 + consumption of fruit and vegetables and sugary drinks (continuous). Time unit: average time difference between two anthropometric questionnaires (approximately 6 months). Sex-specific quartiles of the proportion of UPF intake in the total quantity of food consumed. Cut-offs for quartiles were 10.2, 15.5, and 22.5 for men and 9.9, 15.2, and 22.1 for women.

[a]Estimates β of parameters are interpreted as absolute variation of BMI (Δ BMI).

[b]Interpreted as BMI difference at baseline associated with an absolute increment of 10 in the percentage of UPF in the diet.

[c]Interpreted as BMI gain/time unit when the proportion of UPF in the diet = 0.

[d]Interpreted as BMI gain/time unit associated with an absolute increment of 10 in the percentage of UPF in the diet.

[e]Additional BMI gain/time unit = BMI gain/time unit in quartile 2, quartile 3, or quartile 4, in addition to BMI gain/time unit in Q1.

**Abbreviations:** BMI, body mass index; Q1, quartile 1; SFA, saturated fatty acid; UPF, ultra-processed food

these food groups' consumption in their non–ultra-processed form and increased overweight and obesity risks (P > 0.05), except for products based on meat, fish, or eggs (e.g., unprocessed red and white meat, smoked meats, ham with no added nitrates or additives). In the case of these latter products, the HR for a 100-g increase was 1.16 (1.12–1.20; P < 0.0001) for over-weight, and HR = 1.17 (1.11–1.22; P < 0.001) for obesity.

In secondary analyses, the consumption of unprocessed or minimally processed foods was inversely associated with overweight risk (HR for an absolute increment of 10 in the percent-age of unprocessed/minimally processed foods in the diet = 0.95 [0.92–0.97], P < 0.001), but statistical significance was not reached in obesity analyses (HR = 0.97 [0.94–1.00], P = 0.1).

The associations with overweight and obesity risk were statistically significant in all strata of the population investigated (age groups, subgroups according to sugar and SFA intake, smoking status) except in men, probably due to weaker statistical power (Table A in S1

**Table 3. Associations between UPF intake and risks of overweight and obesity from Cox proportional hazard models, NutriNet-Santé cohort, 2009–2019.**

| Overweight | Q1 | Q2 | Q3 | Q4 | | Continuous[b] | |
|---|---|---|---|---|---|---|---|
| | HR | HR (95% CI) | HR (95% CI) | HR (95% CI) | $P_{trend}$ | HR (95% CI) | *P* |
| *N* cases/non-cases | 1,666/12,092 | 1,706/12,054 | 1,830/11,930 | 1,861/11,898 | | 7,063/47,974 | |
| Model 1 | 1 | 1.06 (1.00–1.14) | 1.19 (1.11–1.28) | 1.26 (1.18–1.35) | <0.001 | 1.11 (1.08–1.14) | <0.001 |
| Model 2 | 1 | 1.07 (1.00–1.14) | 1.19 (1.12–1.28) | 1.30 (1.21–1.39) | <0.001 | 1.11 (1.08–1.14) | <0.001 |
| Model 3 | 1 | 1.06 (0.99–1.13) | 1.18 (1.10–1.26) | 1.24 (1.16–1.33) | <0.001 | 1.10 (1.08–1.13) | <0.001 |
| Model 4 | 1 | 1.05 (0.98–1.13) | 1.17 (1.09–1.25) | 1.22 (1.14–1.31) | <0.001 | 1.10 (1.07–1.13) | <0.001 |
| Model 5 | 1 | 1.05 (0.98–1.13) | 1.17 (1.09–1.25) | 1.22 (1.13–1.31) | <0.001 | 1.10 (1.07–1.13) | <0.001 |
| **Obesity** | Q1 | Q2 | Q3 | Q4 | | **Continuous** | |
| | HR | HR (95% CI) | HR (95% CI) | HR (95% CI) | $P_{trend}$ | HR (95% CI) | *P* |
| *N* cases/non-cases | 687/17,280 | 723/17,245 | 803/17,166 | 853/17,114 | | 3,066/68,805 | |
| Model 1 | 1 | 1.05 (0.94–1.16) | 1.10 (1.00–1.22) | 1.15 (1.04–1.28) | 0.005 | 1.09 (1.05–1.13) | <0.001 |
| Model 2 | 1 | 1.09 (0.98–1.21) | 1.26 (1.13–1.39) | 1.41 (1.27–1.57) | <0.001 | 1.19 (1.15–1.23) | <0.001 |
| Model 3 | 1 | 1.05 (0.95–1.17) | 1.11 (1.00–1.23) | 1.16 (1.05–1.30) | 0.003 | 1.10 (1.06–1.14) | <0.001 |
| Model 4 | 1 | 1.06 (0.95–1.18) | 1.12 (1.01–1.24) | 1.20 (1.08–1.33) | <0.001 | 1.11 (1.07–1.15) | <0.001 |
| Model 5 | 1 | 1.05 (0.95–1.17) | 1.11 (1.00–1.23) | 1.15 (1.03–1.28) | 0.009 | 1.10 (1.05–1.14) | <0.001 |

Qi [i = 1–4] = Quartile, *n* = 55,307 for overweight analyses and 71,871 for obesity analyses. Model 1 was a multi-adjusted Cox proportional hazard model adjusted for age (timescale), sex, educational level (<high school, <2 years after school, ≥2 years after high school), marital status (living alone or not), baseline BMI (continuous), physical activity (high, moderate, low), smoking status (never, former, current), alcohol intake (continuous), number of 24-hour dietary records (continuous), and energy intake (continuous); model 2 = model 1 unadjusted for baseline BMI; model 3 = model 1 + intakes of sodium, sugar, SFAs, and dietary fibre (continuous); model 4 = model 1 + healthy and Western dietary patterns (continuous); model 5 = model 1 + consumption of fruit and vegetables, and sugary drinks (continuous).
[a]Cut-offs for quartiles were 9.9, 14.9, and 21.5 for men and 9.6, 14.5, and 21.1 for women in the overweight analyses; and 9.8, 14.8, and 21.2 for men and 9.6, 14.5, and 21.1 for women in the obesity analyses.
[b]HR for an absolute increment of 10 in the percentage of UPF in the diet.
**Abbreviations:** BMI, body mass index; HR, hazard ratio; SFA, saturated fatty acid; UPF, ultra-processed food

Appendix). The findings remained robust throughout all sensitivity models (e.g., exclusion of cases occurring during the first 2 years and further adjustments).

## Discussion

In this large prospective cohort, participants consuming more UPFs tended to present higher BMI increase during follow-up and had increased risk of becoming overweight and obese, independently of their baseline BMI. These associations remained statistically significant after adjusting for a wide range of socioeconomic and lifestyle factors, and after further adjustments for several indicators of the nutritional quality of the diet.

The increased weight gain in participants consuming more UPFs observed in our study is consistent with several epidemiological studies. A recent study showed that higher consumers of UPFs had a higher risk of weight gain [41]. Moreover, in a recent randomized controlled trial [34], Hall and colleagues allocated participants either to an ultra-processed or minimally processed diet for 2 weeks immediately followed by the alternate diet for 2 weeks. The ultra-processed diet led to an increased energy intake (+508 ± 106 kcal/d), which was highly corre-lated with weight gain (0.8 ± 0.3 kg [*P* = 0.01]), versus a weight loss of 1.1 ± 0.3 kg during the unprocessed diet.

We also observed an increased risk of overweight and obesity in participants consuming more UPFs. Several national cross-sectional studies have shown positive associations between UPF consumption and BMI [36,38]. Furthermore, these results are is in line with 2 ecological

studies suggesting that increased purchases and household availability of UPF were associated with higher BMI and higher obesity prevalence [37,55]. Prospective studies undertaken in Spain [40] and Brazil [41] showed increased risk of overweight/obesity linked to higher UPF intake of a magnitude similar to what we found in our study ($HR_{Q4vs.Q1}$ = 1.26 [1.10–1.45] for the Spanish study and $HR_{Q4vs.Q1}$ = 1.20 [1.03–1.40] for the Brazilian study). The latter study found no association with obesity risk, but it only included overweight participants at baseline and had a shorter follow-up period (3.8 years versus 5.0 years for our study) and participants were older (51.3 years old versus 43.1 for our study). These studies used the contribution of UPFs to daily energy intake, whereas we used the contribution of UPFs to daily quantity of food intake; comparison with our findings is therefore not straightforward.

The positive association observed between UPFs and weight gain may be partly explained by their poorer nutritional quality. Indeed, on average, UPFs tend to be higher in saturated fats, sugar, and energy and poorer in dietary fibre [13–15,17–24], i.e., nutritional factors known to favour obesity onset [5]. However, it is important to note that all analyses were adjusted for daily energy intake, and results remained statistically significant after adjustment for multiple nutritional parameters (key nutrients and food groups, dietary patterns). Therefore, the poor nutritional quality of these foods does not appear to be entirely responsible for the observed association with weight gain, suggesting that non-nutritional bioactive compounds or factors within UPFs may also contribute to explain the findings. First, food processing may affect the food structure and, for two foods with the same nutritional composition, lead to different health impacts [56]. Indeed, the food matrix can influence nutrient and other bioactive food component delivery and bioavailability as well as gut microbiota profile and integrity [56], potentially leading to weight gain in case deleterious nutrients are delivered faster from the food matrix. Second, some food additives, which are specific of UPFs, might be involved in obesity aetiology. For example, saccharin, an artificial sweetener, could potentiate glucose-stimulated insulin release from isolated pancreatic β-cells [57], leading to insulin resistance and potentially weight gain. Some emulsifiers (carboxymethyl cellulose and polysorbate-80, used in >1,500 UPFs in France) induced metabolic perturbations, alterations to the gut microbiota, and low-grade inflammation in mice [58]. Carrageenan, a thickening and stabilizing agent, used in >5,500 products in France and in the top-20 used additives, might increase insulin resistance and inhibit insulin signalling in mouse liver and human HepG2 cells [59,60], which might, in turn, induce weight gain [61]. However, as for most additives, human data on long-term health impacts are still lacking, and potential cocktail effects remain largely unknown. The Europe-funded Additives program will allow us to advance knowledge in this field in the near future [62]. Third, trans fatty acids found in UPFs containing hydrogenated oils have been associated with cardiovascular disease [63] and obesity [64] probably by altering nutrient handling in liver, adipose tissue, and skeletal muscle [65]. Fourth, the long shelf-life of most UPFs increases the risk of contamination from plastic packaging by substances such as bisphenols (which have endocrine-disrupting properties [66]) or phthalates (which are associated with dysregulated sex hormones, obesity, and insulin resistance [67]). A recent study conducted in the US showed that UPF consumption was associated with increased exposure to phthalates [68], which have suggested associations with obesity, especially in children [69]. Lastly, acrylamide, a neo-formed compound created during thermal processing of food as a result of the Maillard reaction, was found to induce adipocyte differentiation and obesity in mice [70]. Furthermore, the increased availability, accessibility, and affordability of these UPFs on the market, in addition to their excessive marketing, might play at least a partial role in these associations, as they contribute to an obesogenic environment [8,71,72], even though we adjusted for energy intake. Therefore, it is important to act on slowing down the obesity

epidemic not only through dietary guidelines but also by ensuring the availability of healthy products and the establishment of an environment that encourages healthy behaviours.

While obesity is a health outcome itself (studied as such in the present analysis), it is also a major risk factor of metabolic diseases such as coronary heart disease, ischemic stroke, and type 2 diabetes [1] but also many cancers [4,5]. Studying the associations between UPF consumption and obesity is relevant as it is, given the metabolic and economic burdens of obesity; but it is also relevant given the role of obesity as a risk factor for these other chronic diseases. In one of our previous prospective studies, we found that UPF consumption was associated with an increased risk of type 2 diabetes [29] and that this association was not fully explained by weight gain. We also found associations between UPF consumption and overall cancer risk, and CVD risk in non-obese participants [24,27]. UPF could therefore play both a direct role in the development of CVD, cancer, and T2D, as discussed elsewhere, but also an indirect role through weight gain and obesity as we observe in the present study. This double pathway is already known and well described for other dietary factors, such as fruit and vegetables, a protector against head and neck cancers with a probable level of evidence according to the WCRF [5] through both a direct effect (i.e., DNA methylation, redox status) and an indirect effect through decreasing overweight and obesity risks [5].

This study has some limitations. First, as it is generally the case in volunteer-based cohorts, participants in the NutriNet-Santé cohort were more often women, with health-conscious behaviours and higher socioeconomic position and educational levels than the general French population [73], and with healthier dietary patterns [74]. We might therefore have underestimated the studied associations due to a lower contrast between extreme quartiles of UPF consumption. Second, misclassifications in the NOVA categories cannot be totally excluded in spite of highly detailed food item lists and consensus reached between 8 scientists. However, this would have led to a nondifferential measurement error (in cases and non-cases), probably biasing results towards the null hypothesis. Moreover, despite adjustment for an extended range of cofactors and stratified analyses, residual confounding cannot be entirely ruled out, thus caution is needed when making causal inference. Furthermore, BMI was used to evaluate overweight and obesity: even though it is validated by the World Health Organization as a detection tool for overweight and obesity, it could be subject to misclassification depending on age, sex, and fat repartition [75,76]. Recently, better estimation tools have been proposed, such as relative fat mass, enabling better prediction of adiposity but requiring clinical waist circumference measurement [77], which was not available prospectively in the cohort and therefore could not be used in the present analysis. Moreover, multiple non-consecutive dietary records for the same individual are considered a valid tool to assess the usual diet [78,79]. We chose to focus on baseline records in this analysis in order to comply with the prospective design and ensure that exposure (UPF consumption) preceded the outcome (BMI change or incident obesity/overweight). This design limits the risk of reverse causality, wherein participants would modify their dietary intakes due to their weight change. However, it cannot be excluded that weight changes occurring late during follow-up may be related to later dietary changes rather than to baseline diet; this probably tended to decrease the strength of the associations observed in this study. Last, the ultra-processed category covers diverse products; this exploratory approach was not designed to focus on a specific food category or to isolate a particular process/additive but has rather allowed us to explore overall exposure to UPF and to observe associations with weight gain, potentially resulting from cumulative intakes and cocktail effects of their ingredients.

Political discussions are currently ongoing in France and in Europe to decrease the number of authorized food additives. Therefore, this study contributes to the mounting level of evidence on food processing and human health needed by public policies to update dietary

guidelines in the future, by integrating aspects of food processing as well as potentially more tightly regulating the policies related to food additives once high-quality results become available.

In conclusion, the results of this large-scale prospective study based on detailed and validated dietary data highlight positive associations between the dietary contribution of UPF with weight gain and risks of overweight and obesity. These associations may be partly explained by the nutritional profile of UPF, but some other dimensions specific to food processing (e.g., food matrix modification, particular food additives and contact materials, neo-formed contaminants) probably also play a key role. Further epidemiological and toxicological research is needed to better understand the underlying mechanisms. The accumulation of consistent findings on the link between UPF and health, which this study contributes to, along with the environmental non-sustainability of these products [9], is leading national public health authorities—in France, e.g. [80]—to recommend privileging the consumption of unprocessed/minimally processed foods and limiting the consumption of UPF. The French National Programme for Nutrition and Health (PNNS) has set a target of a 20% reduction of consumption of UPFs in France by 2022. In addition, these findings might help physicians and dietitians in clinical practice by providing evidence about the role of UPF in weight gain and obesity management.

## Supporting information

**S1 STROBE Checklist. STROBE, Strengthening the Reporting of Observational Studies in Epidemiology**
(DOCX)

**S1 Appendix. Supplementary methods, checking for models' assumptions, and sensitivity analyses.**
(DOCX)

## Acknowledgments

The authors warmly thank all the volunteers of the NutriNet-Santé cohort. We also thank Younes Esseddik, Thi Hong Van Duong, Régis Gatibelza, and Jagatjit Mohinder (computer scientists); Cédric Agaesse (dietitian); Fabien Szabo de Edelenyi, PhD, Julien Allègre, Nathalie Arnault, and Laurent Bourhis (data-managers/biostatisticians); Fatoumata Diallo, MD, Roland Andrianasolo, MD, and Sandrine Kamdem, MD (physicians); and Nathalie Druesne-Pecollo, PhD (operational coordinator) for their technical contribution to the NutriNet-Santé study.

## Author Contributions

**Conceptualization:** Marie Beslay, Bernard Srour, Serge Hercberg, Pilar Galan, Carlos A. Monteiro, Emmanuelle Kesse-Guyot, Chantal Julia, Mathilde Touvier.

**Formal analysis:** Marie Beslay, Bernard Srour.

**Funding acquisition:** Serge Hercberg, Pilar Galan, Mathilde Touvier.

**Investigation:** Marie Beslay, Bernard Srour, Caroline Méjean, Benjamin Allès, Eloi Chazelas, Serge Hercberg, Carlos A. Monteiro, Valérie Deschamps, Giovanna Calixto Andrade, Emmanuelle Kesse-Guyot, Chantal Julia, Mathilde Touvier.

**Methodology:** Marie Beslay, Bernard Srour, Benjamin Allès, Thibault Fiolet, Charlotte Debras, Mélanie Deschasaux, Méyomo Gaelle Wendeu-Foyet, Serge Hercberg, Pilar Galan, Carlos A. Monteiro, Emmanuelle Kesse-Guyot, Chantal Julia, Mathilde Touvier.

**Project administration:** Serge Hercberg, Mathilde Touvier.

**Supervision:** Bernard Srour, Emmanuelle Kesse-Guyot, Chantal Julia, Mathilde Touvier.

**Validation:** Serge Hercberg, Giovanna Calixto Andrade, Chantal Julia, Mathilde Touvier.

**Visualization:** Bernard Srour, Caroline Méjean, Benjamin Allès, Thibault Fiolet, Charlotte Debras, Eloi Chazelas, Mélanie Deschasaux, Méyomo Gaelle Wendeu-Foyet, Pilar Galan, Carlos A. Monteiro, Valérie Deschamps, Emmanuelle Kesse-Guyot, Chantal Julia, Mathilde Touvier.

**Writing – original draft:** Marie Beslay, Bernard Srour.

**Writing – review & editing:** Bernard Srour.

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
