## [Editor Report · Decision Letter 0]

1 Mar 2020

Dear Dr Srour, 

Thank you for submitting your manuscript entitled "Ultra-processed food intake and weight change, overweight and obesity - Findings from the prospective NutriNet-Santé cohort" for consideration by PLOS Medicine.

Your manuscript has now been evaluated by the PLOS Medicine editorial staff and I am writing to let you know that we would like to send your submission out for external peer review.

Kind regards,

Helen Howard, for Clare Stone PhD 

Acting Editor-in-Chief

PLOS Medicine 

plosmedicine.org

---

## [Decision Letter · Decision Letter 1]

9 May 2020

Dear Dr. Srour,

Thank you very much for submitting your manuscript "Ultra-processed food intake and weight change, overweight and obesity - Findings from the prospective NutriNet-Santé cohort" (PMEDICINE-D-20-00661R1) for consideration at PLOS Medicine. 

[LINK]

In light of these reviews, I am afraid that we will not be able to accept the manuscript for publication in the journal in its current form, but we would like to consider a revised version that addresses the reviewers' and editors' comments. Obviously we cannot make any decision about publication until we have seen the revised manuscript and your response, and we plan to seek re-review by one or more of the reviewers. 

We expect to receive your revised manuscript by Jun 01 2020 11:59PM. Please email us (plosmedicine@plos.org) if you have any questions or concerns.

We look forward to receiving your revised manuscript. 

Sincerely, 

Emma Veitch, PhD

PLOS Medicine

On behalf of Clare Stone, PhD, Acting Chief Editor,

PLOS Medicine

plosmedicine.org

*In the abstract (below), suggest clarify the meaning of the figures in brackets - assume these are 95%CI but that should be stated. "UPF intake was associated with a higher risk of overweight (n = 7,063 overweight participants; hazard ratio for an absolute increase of 10% of ultra-processed foods in the diet = 1.11 (1.08-1.14); P<0.0001) and obesity (n = 3,066 incident obese participants; HR10% = 1.09 (1.05-1.13); P<0.0001)." 

*In the last sentence of the Abstract Methods and Findings section, please summarise any key main limitation(s) of the study's methodology (currently there is a good detailed description of this in the main text).

*At this stage, we ask that you include a short, non-technical Author Summary of your research to make findings accessible to a wide audience that includes both scientists and non-scientists. The Author Summary should immediately follow the Abstract in your revised manuscript. This text is subject to editorial change and should be distinct from the scientific abstract. Please see our author guidelines for more information: https://journals.plos.org/plosmedicine/s/revising-your-manuscript#loc-author-summary

*If possible, please reformat in-text reference callouts to use numbers in square brackets (this should be quick and easy if referencing software was used). Many thanks

*Main text, main sections should be Introduction, Methods (not Materials and Methods), Results, Discussion.

*Did your study have a prospective protocol or analysis plan? Please state this (either way) early in the Methods section.

*As the paper reports findings from an observational cohort (prospective), we'd suggest using the STROBE guideline to enhance reporting - please include the completed STROBE checklist as Supporting Information. Please add the following statement, or similar, to the Methods: "This study is reported as per the Strengthening the Reporting of Observational Studies in Epidemiology (STROBE) guideline (SChecklist)." The STROBE guideline can be found here: http://www.equator-network.org/reporting-guidelines/strobe/. When completing the checklist, please use section and paragraph numbers, rather than page numbers.

Comments from the reviewers:

Reviewer #1: Dear editor,

initially I would like to thank you for the invitation to review the manuscript.

This is a cohort study that aimed to investigate the association between the consumption of ultra-processed foods and the risk of overweight and obesity. The article is very well written, clearly presents its relevance and the methods were presented in a detailed way. The authors have considered the possible confounding variables in the adjustment of the models. The results were also clearly exposed. Thus, the publication of the article is recommended because it is a longitudinal population-based study, and there are few studies with this design so far that have analyzed this association between the consumption of ultra-processed foods and the risk of obesity. Only a few points were highlighted that should be further clarified in the text.

Introduction:

line 68: replace today for "nowadays"

Methods:

Did the study population come from all regions of France?

 It is not clear if the consumption data were collected only at baseline (three records) or if these data were obtained every six months as the anthropometric data. Please explain better in the text.

Make clear the age range of study participants. Were adults only?

Results:

table 1: quartile 4 line 6: replace 16680 for 16,680

line 222: "….and with a higher physical activity level." But in table 1 shows that prevalence of high level of PA is higher in the first quartile.

Reviewer #2: I restrict my remarks to statistical aspects of this paper. Although much of what was done is good, I have some issues to resolve before I recommend publication.

Lines 146-152 BMI is a poor measure of obesity. See e.g. my blog post https://medium.com/peter-flom-the-blog/why-bmi-is-a-bad-measure-of-obesity-and-what-is-better-f8a62fc9ca49 Change in BMI at least controls for most of the peoblems. But change in BMi obscures whether the person is obese at all. A small increase in BMI for someone who is normal weight or even underweight may not be problematic. This sort of thing is hard to solve, given the failure to get good day at the start (because good data is hard) but it should be mentioned.

Line 177-178 Don't categorize independent variables In *Regression Modelling Strategies* Frank Harrell lists 11 problems with this and summarizes "Nothing could be more disastrous". I wrote another blog post https://medium.com/@peterflom/what-happens-when-we-categorize-an-independent-variable-in-regression-77d4c5862b6c showing, graphically, the things that can happen. Leave UPF continuous and use splines to investigate nonlinearitiy

Same for line 183 and 193

Table 1 - delete the p value column. P values are not relevant here, effect size is. With these N, even small differences are sig. (NOTE: It is OK to make a table liuke this with quartiles, just don't use quartiles in analysis. However density plots would be better here.

Peter Flom

Reviewer #3: Brazil, May 6th 2020

Manuscript Title: Ultra-processed food intake and weight change, overweight and obesity - Findings from the prospective NutriNet-Santé cohort

Manuscript number: PMEDICINE-D-20-00661R1

Comments to the Author

Overall, this paper is important to the community of researchers in this general area and has technical quality. The results provide a substantial advance over existing knowledge, with clear implications for public health. However, the way the paper is currently written, with much of the methodological detail and appendix missing, it is difficult to assess the methodological approach. 

Abstract

- Line 45: the number >3500 refers to the number of items of the 24h dietary records or the number of foods in a food composition table as presented at material and methods section in line 130? 

- Line 58-61: How about weight change? include more information about weight change on the conclusion - may be useful.

Introduction 

- Overall, the introduction needs to be revised to incorporate other aspects of the complexity of overweight, obesity and weight change. The authors can then incorporate the literature on food access, for example. There are complex interactions between biological, behavioral, social and environmental factors that are involved in overweight, obesity and weight change; they need to be contemplated.

- The author highlights the issue of obesity as an important risk factor for metabolic diseases. I suggest that the author reflects: "Is consumption of AUP a risk factor for another risk factor? Or a risk factor for a metabolic disease (ICD-10 code E66) that is simultaneously related to other chronic non-communicable diseases? How can the study of obesity as a risk factor minimize the results found here?

- In addition to the lack of prospective studies, explicit the study's contribution to the state of the art on the subject.

Material and methods

- In the methodology, additional details regarding process of recruitment are needed. Is there any possible implications of this selection for the evidence produced by the study? I realize that additional details have already been published but it is hard to understand the process without a few more specific details. 

- In the data collection, can the authors be more specific about the number of waves that the cohort presents, the time difference between them and whether the variable outcome and explanatory were measured in all waves?

- Line 128-129: Can the authors expand why dietary intakes from the baseline 24-hour dietary records were considered as usual dietary intakes? What are the limitations to this premise? 

- Line 131: Can the authors be more specific about the method proposed by Black? What characterizes "under-reporters"? What were these definitions based on? I realize that additional details have already been published but it's hard to understand the exclusion without a few more specific details. And how about over-reporters? They were excluded?

- Line 146: And in the "Obesity and overweight" subsection how was the incidence of overweight and obesity calculated. 

- I suggest a subsection entitled "Weight change" (before or after obesity and overweight) or complete the subsection "Obesity and overweight". How was the weight change assessed? Delta? % weight loss? MBI change as present on Line 182? This is one of the outcomes of this study and its calculation must be detailed. Finally, if this outcome was assessed as a change in BMI, it is important to clarify it in the title and throughout the manuscript.

- 159-160: Can the authors be more specific about the categories of low, moderate and high physical activity levels? It is suggested to insert the cutoff points.

- Include indicators of the nutritional quality of the diet (alcohol, sodium, saturated fatty, dietary fiber and sugar intake, number of dietary records, fruit and vegetables, sugary drinks, Healthy and Western dietary patterns) as covariates. They were presented only in the tables and not detailed in the methods. Please, how were they obtained? 

- I suggest the exclusion of covariates present at that section, but not applied in the statistical analyses or adjustment, for example: time spent for screen watching and sedentary behaviors. 

- Line 165: Can the authors explain why exclude participants with missing anthropometric in place of apply a method of processing these missing data, for example imputation - as done for covariates (Line 178-182)? Do individuals excluded because of this information differ from those included in the study? 

- It isn't clear how sex-specific quartiles were accessed. That detail can lead the reader to better understand how only one data for sex-quartile were presented per quartile on table 1 for example. 

- Can the authors clarify why adjust for sex if the proportion of UPF in the diet was sex-specific quartiles? 

- Can the authors clarify why adjust for Healthy and Western dietary patterns and consumption of fruit and vegetables? Fruit and vegetables consumption do not also refers to a Healthy dietary patterns?

- Improve the writing of the additional adjustments for sugar, fiber, sodium, and saturated fatty acid (SFA). Can the authors expand why that adjust were performed?

- Line 202-2012: Improve the wording of the two last paragraphs of the statistical analyses. It is confusing. Perhaps, highlight that it refers to Secondary and sensitivity analyses.

- Line 202-204: Can the authors clarify why reverse causality is a problem in this study. It is not a prospective? 

Results

- Much of the information in the tables were not included or explained in the Material and Methods section. This should be reorganized. For example, additional BMI gain, Time…

- Figure 1: separate the exclusions of those who were "overweight or obese" from those "missing from anthropometry".

- Table 1: The Sex-specific quartiles of the proportion of ultra-processed food intake in the total quantity of food consumed were accessed for all quartiles, correct? So include letter b for all quartiles or clarify the note for example: Cut-offs for quartiles were 10.2, 15.5 and 22.5 for men and 9.9, 15.2 and 22.1 for women, respectively.

- Table 1 Note: Please add the full IPAQ name and Standard Deviation (SDs). And a brief explanation about IPAQ categories (low, moderate and High). The table must be self-explanatory.

- Table 2 - Standardize the number of decimal places to the p value

- Table 2 Note: Please include the definition of the abbreviation for BMI.

- Table 3 - Standardize the number of decimal places to the p value

- Table 3 Note: Please include the definition of the abbreviation for Q1, Q2, Q3 e Q4.

Discussion

- Exclude line 319, 354 and 374

- 296-300: I suggest rephrasing or deleting these phrases to better fit the population (adults), object of study of the manuscript.

- The presentation of non-nutritional bioactive compounds as a hypothesis for the associations found is fantastic and real. But how these findings might inform policies and/or programming? It is urgent to include aspects of environment on that discussion.

- The discussion would benefit from integrating the findings with additional existing literature about obesogenic environment, which would connect UPF consumption and obesity. Specially on the second paragraph in which the author highlights the nutritional quality of UPF does not appear to be entirely responsible association with weight gain. 

- Line 355: replace "First" for "first".

- The sentences "[…] However, this would have led to a non-differential measurement error (in cases and non-cases), probably biasing results towards the null hypothesis. Moreover, despite adjustment for an extended range of cofactors and stratified analyses, residual confounding cannot be entirely ruled out, thus no causality can be established […]". They are very emphatic. Causality must be established with caution, but not established.

- The author should exclude the self-reported anthropometric as a limitation, once the data has been validated (Lassale C, Péneau S, Touvier M, Julia C, Galan P, Hercberg S, et al. Validity of web-based self551 reported weight and height: results of the Nutrinet-Santé study. J Med Internet Res. 2013 Aug 552 8;15(8):e152)

- The author should include a paragraph with the potential of the study.

- The conclusions would benefit from providing some insight of how these findings might inform policies and/or programming and with broader implications of the study findings.

Reference list

- Add space between the words that make up the name of scientific journals

 #8 Obes Rev.

 #9: Int J Behav Nutr Phys Act.

 #15 and #19: Rev Saude Publica

#17 and #35: Am J Clin Nutr.

#18: Eur J Clin Nutr

#39: Int J Obes Relat Metab

#40 and #42: Br J Nutr.

#41: J Acad Nutr Diet.

#48: Int J Public Health

#49: Med Sci Sports Exerc.

-#42 and #43 are the same. Exclude one of them and review citations numbers in the manuscript.

[LINK]

---

## [Decision Letter · Decision Letter 2]

23 Jun 2020

Dear Dr. Srour,

Thank you very much for re-submitting your manuscript "Ultra-processed food intake in association with BMI change and risk of overweight and obesity - Findings from the prospective NutriNet-Santé cohort" (PMEDICINE-D-20-00661R2) for review by PLOS Medicine.

I have discussed the paper with my colleagues and the academic editor and it was also seen again by some of the original reviewers. I am pleased to say that provided the remaining editorial and production issues are dealt with we are planning to accept the paper for publication in the journal.

[LINK]

We look forward to receiving the revised manuscript by Jun 30 2020 11:59PM. 

Sincerely,

Clare Stone, PhD

Managing Editor 

PLOS Medicine

plosmedicine.org

Requests from Editors:

Can I clarify the research design, please? I suspect that the data have been collected prospectively, but that this analysis is retrospective - you use the word "prospective" twice in the abstract. Please can be clear about the design. 

Title- please change from 

Ultra-processed food intake in association with BMI change and risk of overweight and obesity - Findings from the prospective NutriNet-Santé cohort

To 

Ultra-processed food intake in association with BMI change and risk of overweight and obesity; a prospective analysis of the French NutriNet-Santé cohort

Please add aggregate demographic details to go in the abstract; adjustment factors should be listed

- p<0.0001 -> p<0.001 please

THE STROBE – can I just query the symbols used. Noting that sections and paragraphs should be used. 

Comments from Reviewers:

Reviewer #2: The authors have addressed my concerns and I now recommend publication.

Peter Flom

Reviewer #3: Comments to the Author

Great job of responding to feedback. The manuscript still needs additional proofreading. Here are minor additional comments.

Introduction 

Thanks for your explanation about obesity as a risk factor for metabolic disease. I think it's valuable, though the writing needs also presents obesity as a metabolic disease (ICD-10 code E66). The results can be maximized from that perspective. Presenting obesity as a disease is an important step in its management. It is important to incorporate this discourse in our work on the topic.

The first paragraph is long. I suggest that there are two paragraphs starting from line 110, which separates epidemiology from risk factors.

Line 115-121 - thanks for adding in this paragraph. I think it's valuable, though the writing needs a more recent or a classic citation/reference for statement about food system and built environment. We suggest: (1) Swinburn BA, Kraak VI, Allender S, Atkins VJ, Baker PI, Bogard JR, et al. The Global Syndemic of Obesity, Undernutrition, and Climate Change: The Lancet Commission report. The Lancet. 2019;393: 791-846. doi:10.1016/S0140-6736(18)32822-8; (2) Swinburn, B., Egger, G., Raza, F. Dissecting obesogenic environments: the development and application of a framework for identifying and prioritizing environmental interventions for obesity. Prev. Med., v.29, n.6, p.563-70, 1999.

Material and methods

Line 216-220 - insert the unit of measurement of the covariates: sugar, fibre, sodium, and saturated fatty acid (SFA), fruit, vegetables, and sugary drinks.

Line 223-228 - why italics?

Discussion

In the last paragraph, focus on France and exclude issues related to other countries like Brazil.

[LINK]

---

## [Editor Report · Decision Letter 3]

22 Jul 2020

Dear Dr. Srour, 

On behalf of my colleagues and the academic editor, Dr. Aline Cristine Souza Lopes, I am delighted to inform you that your manuscript entitled "Ultra-processed food intake in association with BMI change and risk of overweight and obesity; a prospective analysis of the French NutriNet-Santé cohort" (PMEDICINE-D-20-00661R3) has been accepted for publication in PLOS Medicine. 

PRODUCTION PROCESS

PRESS

PROFILE INFORMATION

Thank you again for submitting the manuscript to PLOS Medicine. We look forward to publishing it. 

Best wishes, 

Clare Stone, PhD

Managing Editor 

PLOS Medicine

plosmedicine.org